# evalhyd v0.1.2: a polyglot tool for the evaluation of deterministic and probabilistic streamflow predictions

Thibault Hallouin[1,a], François Bourgin[1], Charles Perrin[1], Maria-Helena Ramos[1], and Vazken Andréassian[1]

[1]Université Paris-Saclay, INRAE, HYCAR, Antony, France
[a]now at: French Geological Survey (BRGM), Water Resources Unit, Orléans, France

**Correspondence:** Thibault Hallouin (thibault.hallouin@inrae.fr)

**Abstract.** The evaluation of streamflow predictions forms an essential part of most hydrological modelling studies published in the literature. The evaluation process typically involves the computation of some evaluation metrics, but it can also involve the preliminary processing of the predictions as well as the subsequent processing of the computed metrics. In order for published hydrological studies to be reproducible, these steps need to be carefully documented by the authors. The availability of a single tool performing all of these tasks would simplify the documentation by the authors, but also the reproducibility by the readers. However, this requires for such a tool to be polyglot (i.e. usable in a variety of programming languages) and openly accessible, so that it can be used by everyone in the hydrological community. To this end, we developed a new tool named `evalhyd` that offers metrics and functionalities for the evaluation of deterministic and probabilistic streamflow predictions. It is open source and it can be used in Python, in R, in C++, or as a command line tool. This article describes the tool and illustrates its functionalities using Global Flood Awareness System (GloFAS) reforecasts over France as an example data set.

## 1 Introduction

Whether it is referred to as validation, evaluation, or verification (Beven and Young, 2013, sect. 5), the action of comparing streamflow model outputs against streamflow observations is routinely performed by hydrological modellers. This comparison is typically carried out in view to estimate model parameters or to assess model performance. To these ends, one or more measures of the goodness of fit between streamflow time series is used, sometimes referred to as objective functions, performance metrics, or verification scores depending on the context. While there is a variety of metrics to perform such a task (Crochemore et al., 2015; Anctil and Ramos, 2017; Huang and Zhao, 2022), those that are chosen are often the same, for instance the Nash-Sutcliffe Efficiency (NSE) (Nash and Sutcliffe, 1970) or the Kling-Gupta Efficiency (KGE) (Gupta et al., 2009) for the deterministic evaluation of streamflow predictions, or the Brier Score (BS) (Brier, 1950) or the Continuous Rank Probability Score (CRPS) (Hersbach, 2000) for the probabilistic evaluation of streamflow predictions. Note that we use the term predictions as an umbrella term to include both simulations and forecasts (see definitions in Beven and Young, 2013, sect. 3). Moreover, before computing the metrics, streamflow time series are often subject to some preliminary processing (e.g.

handling of missing data, data transformation, selection of events) and, after computing them, the metrics can also be subject to some subsequent processing (e.g. sensitivity analysis, uncertainty estimation).

The computation of the metrics described in the literature is sometimes performed directly as part of the workflow analysing the predictions, which is error-prone and hardly traceable unless carefully explained in the publication. In other cases, specific tools dedicated to the computation of the metrics are used, but they can be private or under commercial licensing, which limits their accessibility; they are not always open source, which limits their transparency; or they rely on a specific programming language (e.g. Python, R, Matlab, Julia), which limits their universality. The existence of different tools is unavoidable when these

are language-specific. This may lead to discrepancies in the computation of the metrics, or in the preliminary and subsequent processing steps inherent to the evaluation workflow. Together, these discrepancies are likely impeding on the reproducibility of published results in hydrological sciences (Hutton et al., 2016; Stagge et al., 2019).

Unlike meteorological predictions, which are typically issued on a spatial grid, hydrological predictions are often issued at discrete locations along the river network typically, but not only, where hydrometric stations are located. In addition,

hydrological predictions often focus on specific extreme events (i.e. floods, droughts). In hydrology, evaluation tools have thus to be adapted to these situations. There exists a variety of evaluation tools with varying degrees of adequacy with the specificities of hydrological predictions. Inventories of tools specific to hydrology can be found in Slater et al. (2019) and at https://cran.r-project.org/view=Hydrology (last access: 23 June 2023) for R, or at https://github.com/raoulcollenteur/ Python-Hydrology-Tools (last access: 23 June 2023) for Python. The Ensemble Verification System (EVS) (Brown et al., 2010)

is certainly the most advanced example of an evaluation tool dedicated to discrete hydro-meteorological ensemble predictions. But one major drawback of EVS is that it requires for the inputs to be provided as data files of a certain non-standardised format. This requires reformatting model outputs, which can be truly limiting and inefficient when dealing with large sets of model outputs (e.g. large model ensembles, or large samples of catchments). In addition, the use of a non-standardised file format limits its integration in reproducible analysis workflows (Knoben et al., 2022). Other evaluation tools have been developed,

mostly from the meteorological community, e.g. `verification` (https://cran.r-project.org/web/packages/verification, last access: 10 May 2023) or `scoringRules` (https://cran.r-project.org/web/packages/scoringRules, last access: 10 May 2023) in R, `ensverif` (https://pypi.org/project/ensverif, last access: 10 May 2023) or `properscoring` (https://pypi.org/project/ properscoring, last access: 10 May 2023) in Python. However, these are not specific to hydrology, they lack preliminary and subsequent processing aspects, or they feature a limited diversity of metrics.

In this context, an evaluation tool that is tailored to hydrology, both in terms of the richness in the metrics it gives access to and the relevance of the functionalities it features, and that can be used in a variety of programming languages has the potential to offer a community-wide solution to improve on the reproducibility of published hydrological studies. Indeed, the packaging of all these aspects into a single tool contributes to lessening the need for providing detailed explanations in scientific publications while guaranteeing the ability for readers and reviewers alike to perform the same processing and

computations, and to apply the same hypotheses. This article presents `evalhyd`, an evaluation tool specifically designed to meet such needs in hydrological evaluation. In particular, it is freely accessible (no commercial licensing), it is transparent (open source), it is efficient (compiled core), and it is universal (Python, R, C++, command line interfaces). In addition, it

features advanced methods to evaluate hydrological predictions, such as data stratification, bootstrapping, and multivariate scoring which contribute to advancing and harmonising best practices in hydrological evaluation.

This article first describes the objectives of the tool, before its design principles, its main functionalities, and its available evaluation metrics are unfolded in turn. Then, a case study using the tool is provided as an illustration of its capabilities. Finally, the limitations and the perspectives for further developments of the tool are discussed.

## 2  Objectives of the tool

The purpose of `evalhyd` is to provide a utility to evaluate streamflow predictions. It aims to feature the most commonly used metrics for deterministic and probabilistic evaluation (see e.g. Huang and Zhao, 2022), as well as all the necessary functionalities required to preliminarily process the data analysed and subsequently process the metrics computed. Typically, these are rarely documented in the literature and this can limit the reproducibility of research findings (Hutton et al., 2016).

In line with Principle 3 for Open Hydrology advocated for by Hall et al. (2022), the tool must be open access. The tool must also be polyglot, which means that it must be usable in a variety of open source programming languages, in order to be usable by as many users as possible. In practice, this means that a separate package must be available for each programming language, thus forming a software stack.

## 3  Design principles

### 3.1  A compiled core with thin bindings

The software stack, named `evalhyd`, features a core library written in C++, named `evalhyd-cpp`, which implements all the available functionalities and evaluation metrics. The implementation of the core is written in a compiled language to be computationally efficient. This is important since large-sample studies (see e.g. Gupta et al., 2014) and large hydro-meteorological ensembles (see e.g. Schaake et al., 2007) have become very common in hydrological model evaluation studies. In fact, `evalhyd` has already been applied to large-sample studies on multi-model approaches (Thébault, 2023).

In addition, the stack features bindings which are distinct packages whose only purposes are to interface the core with high-level programming languages. These are named `evalhyd-python` for Python and `evalhyd-r` for R. They are intended to be as thin as wrappers can be to avoid duplicated efforts across bindings. The core library is also directly usable in C++ as a header-only library. A command line interface, named `evalhyd-cli`, also exists for those not using any of the programming languages mentioned. The only drawback with the latter is that it is only able to work via intermediate data files, which presents the same limitations as EVS of relying on a specific data file format. On the contrary, the Python and R bindings work directly with native data structures of those languages, thus completely eliminating the need to decide on a specific file format, and allowing for any model output to be processed without requiring for them to be reformatted.

Note that both the core library and the bindings use the libraries from the `xtensor` software stack underneath (https://github.com/xtensor-stack, last access: 12 May 2023) and, in particular, they leverage the bindings available for multiple

**(a)** C++ interface

```cpp
#include <xtensor/xtensor.hpp>
#include <evalhyd/evald.hpp>
xt::xtensor<double, 2> obs =
{{4.7, 4.3, 5.5, 2.7, 4.1}};
xt::xtensor<double, 2> prd =
{{5.3, 4.2, 5.7, 2.3, 3.1},
{4.3, 4.2, 4.7, 4.3, 3.3},
{5.3, 5.2, 5.7, 2.3, 3.9}};
auto res = evalhyd::evald(obs, prd, {"NSE"});
```

**(b)** Python interface

```python
import numpy
import evalhyd
obs = numpy.array([[4.7, 4.3, 5.5, 2.7, 4.1]])
prd = numpy.array([[5.3, 4.2, 5.7, 2.3, 3.1],
[4.3, 4.2, 4.7, 4.3, 3.3],
[5.3, 5.2, 5.7, 2.3, 3.9]])
res = evalhyd.evald(obs, prd, ["NSE"])
```

**(c)** R interface

```r
library(evalhyd)
obs ← rbind(c(4.7, 4.3, 5.5, 2.7, 4.1))
prd ← rbind(c(5.3, 4.2, 5.7, 2.3, 3.1),
c(4.3, 4.2, 4.7, 4.3, 3.3),
c(5.3, 5.2, 5.7, 2.3, 3.9))
res ← evalhyd::evald(obs, prd, c("NSE"))
```

**(d)** Command line interface

```
1   cat "./obs.csv"
    4.7,4.3,5.5,2.7,4.1
2   cat "./prd.csv"
    5.3,4.2,5.7,2.3,3.1
    4.3,4.2,4.7,4.3,3.3
    5.3,5.2,5.7,2.3,3.9
3   res=$(evalhyd evald "./obs.csv" "./prd.csv" "NSE")
```

**Figure 1.** Comparison of the interfaces for the deterministic entry point `evald` across the `evalhyd` software stack through a simple example evaluating deterministic predictions (`prd`, of shape $\{series : 3, time : 5\}$) against observations (`obs`, of shape $\{1, time : 5\}$) using the Nash-Sutcliffe efficiency (NSE): **(a)** C++ interface fed with `xtensor` data structures, **(b)** Python interface fed with `numpy` data structures, **(c)** R interface fed with R data structures, and **(d)** command line interface fed with data in CSV files.

programming languages (i.e. `xtensor-python` and `xtensor-r`). As such, there is scope for the development of a binding for Julia in the future (i.e. using `xtensor-julia`).

For the remainder of this article, we will keep referring to the software stack as the one tool `evalhyd` for convenience, since all the functionalities and evaluation metrics are available in all packages.

### 3.2 A two-entry-point interface

Across the software stack, the interfaces are intentionally kept as identical as possible given the syntactic differences across languages (as illustrated in Figure 1 and Figure A1). Thus, users of different `evalhyd` packages can easily help each other, and this also minimises inconveniences when users transition from one language to another.

Each package has two entry points to its interface, named `evald` and `evalp`, two functions for the evaluation of deterministic predictions (see trivial examples in Figure 1) and probabilistic predictions (see trivial examples in Figure A1), respectively. Note that the names and the order of the parameters of these two entry points are strictly identical across the software stack.

### 3.3 A multi-dimensional paradigm

Both `evald` and `evalp` entry points take multi-dimensional data sets as inputs in view to accommodate for the specific needs identified for deterministic and probabilistic predictions, as detailed below.

For probabilistic evaluation, streamflow predictions $\mathbf{Q_{prd}}$ are at the minimum two-dimensional, that is $\mathbf{Q_{prd}} \in \mathbb{R}^{M \times T}$ (with dimensions $M$ for the ensemble members, and $T$ for the time steps), corresponding to an ensemble forecast for a given site and a given lead time. However, there are also metrics that need computing on multiple sites or on multiple lead times at once. This is why the predictions are expected to be four-dimensional, that is $\mathbf{Q_{prd}} \in \mathbb{R}^{S \times L \times M \times T}$ (with dimensions $S$ for the sites, $L$ for the lead times, and $M$ and $T$ as previously), and the observations $\mathbf{Q_{obs}}$ to be two-dimensional, that is $\mathbf{Q_{obs}} \in \mathbb{R}^{S \times T}$. Note that the input ranks are kept fixed, even if the problem does not require all dimensions. For example, even if the problem only features one site and one lead time, the predictions must be made four-dimensional.

For deterministic evaluation, streamflow predictions $\mathbf{Q_{prd}}$ are at the minimum one-dimensional, that is $\mathbf{Q_{prd}} \in \mathbb{R}^{T}$, corresponding to a simulation for a given site. However, it seems opportune to accommodate for multiple simulation time series, as is the case in Monte Carlo simulations, deterministic forecasts for multiple lead-times, or multi-model approaches for example. This is why the predictions are expected to be two-dimensional, that is $\mathbf{Q_{prd}} \in \mathbb{R}^{X \times T}$ (with dimension $X$ for the series, that could be e.g. lead times in a forecasting context or samples in a Monte Carlo simulation context, and $T$ as previously), and the observations $\mathbf{Q_{obs}}$ to be two-dimensional, that is $\mathbf{O} \in \mathbb{R}^{1 \times T}$ (with dimension $T$ as previously). Note that for convenience, the Python and R bindings allow for one-dimensional inputs, which can be useful when used in a model parameter estimation context (i.e. $\mathbf{Q_{prd}} \in \mathbb{R}^{T}$ and $\mathbf{Q_{obs}} \in \mathbb{R}^{T}$).

## 4  Key functionalities

### 4.1  Memoisation

Certain evaluation metrics require the same intermediate computations. For example, the Nash-Sutcliffe efficiency (Nash and Sutcliffe, 1970) and the Kling-Gupta efficiency (Gupta et al., 2009) both require to compute the observed variance, so it is computationally more efficient to compute this variance once, and reuse it if both metrics are requested by the user.

In computer science, this refers to the concept of memoisation (or memo functions) first introduced by Michie (1968) in the context of machine learning. `evalhyd` applies this concept by isolating the recurrent intermediate computations across the evaluation metrics, and by storing these for potential later reuse if multiple metrics are requested at once. This is why the user is advised to request all desired metrics in a single call to `evalhyd` rather than in separate calls.

### 4.2  Handling of missing data

Streamflow observations are seldom complete and missing data is very common. Various data filling methods exist to overcome this problem (see e.g. Gao et al. (2018) for a review), but this is typically done by the streamflow data providers themselves where possible, so it is not deemed as a task that an evaluation tool should perform. Therefore, if streamflow observations remain lacunar, `evalhyd` disregards the time steps where observations are missing (i.e. where they are set as *not a number*).

In addition, for the case of probabilistic evaluation, some streamflow prediction values may also need to be flagged as *not a number*. This is illustrated in Figure 2 for a fictitious data set featuring daily predictions for three lead times. By design,

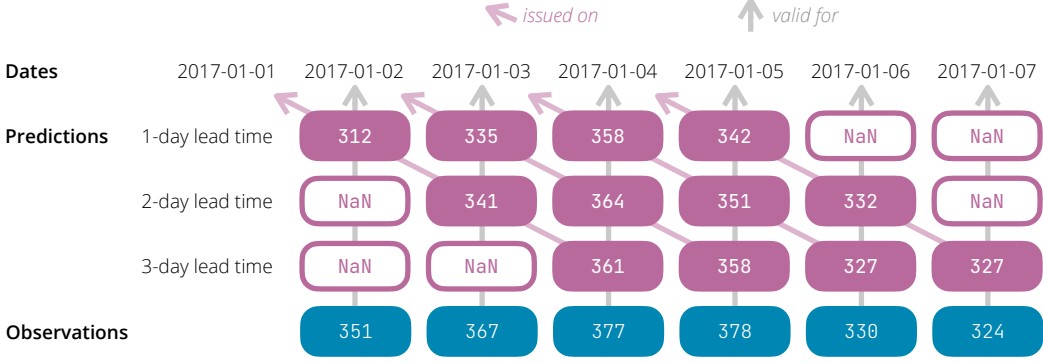

**Figure 2.** Illustration of the need to insert "not a number" values (symbolised as hollow rounded rectangles containing `NaN`) before and/or after the prediction series to align the prediction validity dates (i.e. issue date + lead time) with the observation dates when several lead times are considered at once. This example features a small fictitious situation where daily forecasts are issued on four consecutive days (2017-01-01, 2017-01-02, 2017-01-03, and 2017-01-04) for three lead times (1-day, 2-day, and 3-day). Each filled rounded rectangle contains a fictitious streamflow value (predicted on the first three rows, observed on the last row).

`evalhyd` expects only one observation time series for a given site. Therefore, when several lead times are considered at once, a temporal shift of the predictions must be applied, and observed dates for which a forecast is not made (i.e. where date $\neq$ forecast issue date + lead time) must be identified as *not a number*. That is because the earliest observations are not needed for the longer lead times and the latest observations are not needed for the shorter lead times.

### 4.3 Masking

Depending on the context of the evaluation, it may be desirable to only consider sub-periods of the streamflow records: for example, to focus on specific flood or drought events, or more generally to consider only high or low flows. In the literature, this is sometimes referred to as conditional verification and often used to stratify the evaluation into different meteorological/hydrological conditions in view to diagnose specific physical processes that prevail under particular conditions (Casati et al., 2008, 2022) and to avoid averaging out different forecast behaviours (Bellier et al., 2017). `evalhyd` offers two avenues to perform such subsets: temporal masking and conditional masking.

Temporal masking corresponds to directly providing a mask (i.e. a sequence of boolean values of the same length as the streamflow time series) whose values are set to *true* for those time steps that should be considered in the evaluation, and to *false* otherwise (see Figure 3 for a trivial example illustrating the masking mechanism). This mask is typically expected to be generated by the user in the high-level programming of their choice.

Conditional masking corresponds to a convenient alternative to generate the temporal mask for the user through the specification of (a) condition(s). The condition(s) can be based on streamflow values (either streamflow observations, or mean or median streamflow predictions), or on time indices. Table 1 provides examples of the syntax to use to specify such conditions alongside their plain meaning. The conditions on streamflow values are helpful to focus the evaluation on particular flow ranges

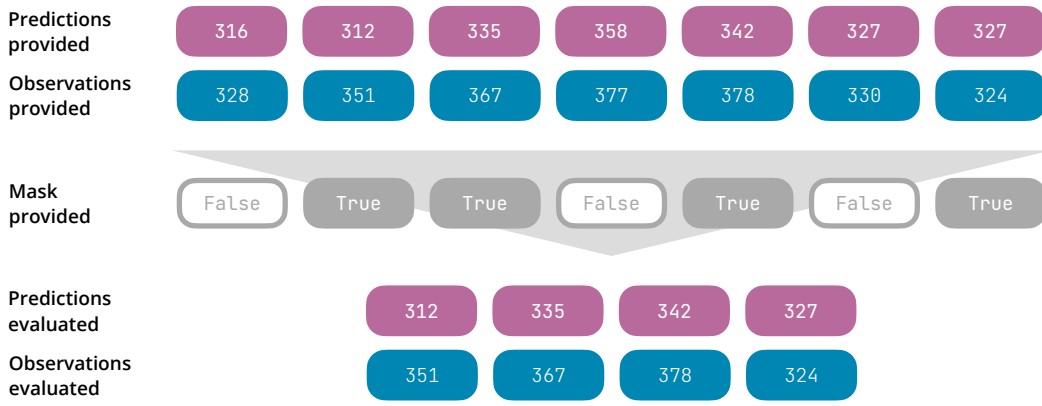

**Figure 3.** Illustration of the temporal masking functionality. The first two rows correspond to the predictions and observations time series provided by the user (each filled rounded rectangle contains a fictitious streamflow value). The third row corresponds to the temporal mask provided by the user as a boolean sequence: it features the same length as the time series, and it contains `True` values at the indices to consider in the evaluation (i.e. 2nd, 3rd, 5th, 7th) and `False` at the indices to ignore in the evaluation (i.e. 1st, 4th, 6th). The result of the masking is displayed on the last two rows, where only the 2nd, 3rd, 5th, and 7th predicted and observed streamflow values have been retained for the evaluation.

(e.g. low flows or high flows) by specifying streamflow thresholds, whereas the conditions on time indices are helpful to focus on specific flow events (e.g. notable floods or droughts) by selecting relevant time steps. Note that the conditional masking

functionality must be employed with care as it can lead to some synthetic bias in the evaluation as explained in the Limitations section of this article.

Note that the concept of memoisation is also applied by `evalhyd` if several masks are provided. Since metrics typically compute some form of error between pairs of observation and prediction time steps before applying some form of temporal reduction (e.g. sum or average), it is computationally more efficient to store the error computed for individual pairs before

reducing them according to the various masks (i.e. subset periods) provided or generated.

### 4.4   Data transformation

It is common practice in hydrology to apply transform functions to the streamflow data prior to the computation of the evaluation metrics. This is typically done because hydrological models tend to produce larger errors on flows of higher magnitude, which results in putting more emphasis on high-flow periods when computing the metric. To reduce the emphasis on high flows

or to change the emphasis altogether, various transform functions can be applied (see e.g. Krause et al., 2005; Oudin et al., 2006; Pushpalatha et al., 2012; Pechlivanidis et al., 2014; Garcia et al., 2017; Santos et al., 2018).

`evalhyd` offers several data transformation functions to be applied to both the streamflow observations and predictions prior to the computation of the evaluation metrics. These include the natural logarithm function, the reciprocal function, the square root function, and the power function. For those functions not defined for zero (i.e. the reciprocal function, the natural

**Table 1.** Example of masking conditions possible in `evalhyd`.

| Condition | Plain meaning |
|---|---|
| **On streamflow values** | |
| `q_obs{>median}` | consider periods where streamflow observations are greater than their median |
| `q_prd_median{<=250}` | consider periods where median streamflow predictions[a] are lower than or equal to 250[b] |
| `q_prd_mean{<qtl0.8}` | consider periods where mean streamflow predictions[a] are lower than their 80th percentile |
| **On time indices** | |
| `t{20:53}` | consider period from 21st[c] time step to 53rd[c] time step[d] |
| `t{12,13,14}` | consider period including 11th[c], 12th[c], and 13th[c] time steps |

[a] conditions on streamflow predictions are only available for probabilistic evaluation

[b] streamflow unit in condition is assumed to be the same as in the input data

[c] indexing starts at zero (i.e. first time step is at index 0)

[d] last index is not included

logarithm, or the power function with a negative exponent), a small value is added to both the streamflow observations and predictions as recommended by Pushpalatha et al. (2012): by default, one hundredth of the mean of the streamflow observations is used, but it can be customised by the user.

This functionality is the only one that is only applicable to deterministic predictions, i.e. via the `evald` entry point, because it is currently only common practice in a deterministic context. Later versions of the tool could remedy this depending on user feedback.

## 4.5 Bootstrapping

It is crucial to assess the sampling uncertainty in the evaluation metrics, that is to say their variability from one study period (i.e. one sample) to another. Clark et al. (2021) recommend using a non-overlapping block bootstrapping method, where blocks are taken as distinct hydrological years to preserve seasonal patterns and intra-annual auto-correlation.

`evalhyd` implements such a bootstrapping functionality to provide an estimation of the sampling uncertainty in the evaluation metrics it computes. The number of samples (i.e. the number of sub-periods drawn from the whole period available in the data) considered as well as the number of blocks (i.e. complete years) in each sample are specified by the user. Then, the tool randomly samples with replacement years within the whole study period accordingly (see Figure 4 for a trivial example illustrating the bootstrapping mechanism). Once again, the concept of memoisation is exploited to avoid performing several times the same computations from one sample to another. As many evaluation metric values are computed as there are samples. These values can either be returned directly or as summary statistics (either mean and standard deviation, or distribution of

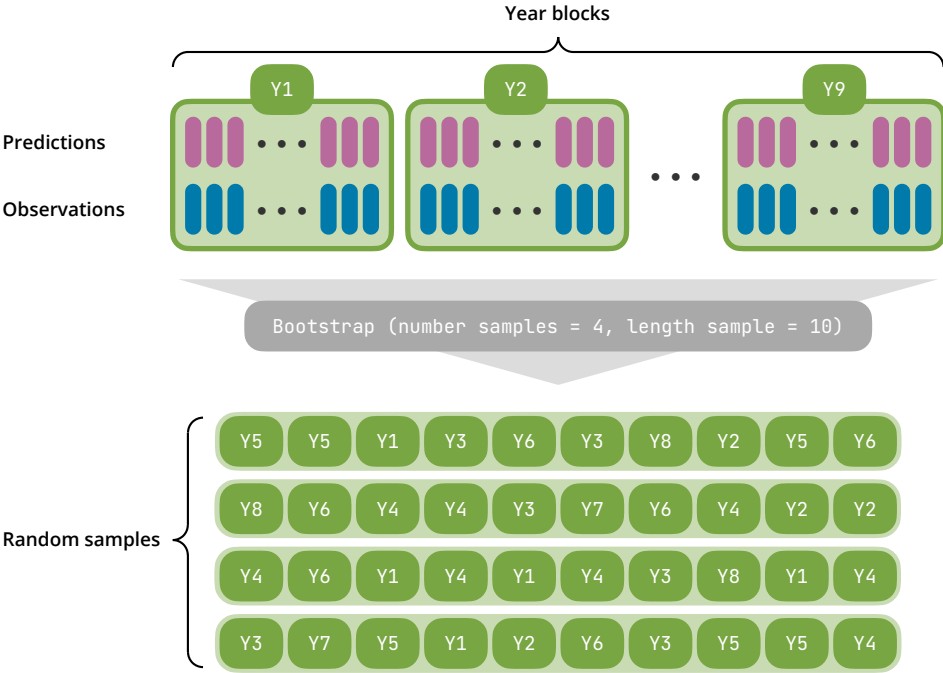

**Figure 4.** Illustration of the bootstrapping functionality. The first two rows correspond to the predictions and observations time series provided by the user (each time step is symbolised by a vertical coloured stick) covering a period of nine years that are sliced into non-overlapping blocks of one year each. The bootstrapping functionality is then applied with two parameters: a number of samples equals to four and a length of ten years for each sample. It produces four synthetic samples made of ten randomly drawn (with replacement) year blocks that are concatenated to form four pairs of ten-year-long time series of predictions and observations.

quantiles). To generate the same samples from one evaluation to another, the seed of the pseudo-random number generator can be fixed by the user.

## 5 Evaluation metrics

The tool features a variety of metrics for the evaluation of deterministic and probabilistic streamflow predictions. The collection of deterministic metrics is presented in Table 2, while the collection of probabilistic metrics is presented in Table 3. The deterministic metrics are accessible via the `evald` entry point, and the probabilistic metrics are accessible via the `evalp` entry point.

Skill scores are sometimes used to compare the score of the predictions with the score of a reference, e.g. Kling-Gupta

Efficiency Skill Score (KGESS) from KGE (Knoben et al., 2019) or Continuous Rank Probability Skill Score (CRPSS) from CRPS (see e.g. Yuan and Wood, 2012). They are formulated as the difference between the predictions score and the reference score divided by the difference between the perfect score and the reference score (see e.g. Wilks, 2011, eq. 8.4). However, the

**Table 2.** Collection of deterministic metrics available via `evald` entry point in `evalhyd`.

| Identifier | Range* | Unit | Details | Related references |
|---|---|---|---|---|
| MAE | $[\mathbf{0}, +\infty)$ | same as q | Mean absolute error | Willmott and Matsuura (2005); Moriasi et al. (2007) |
| MARE | $[\mathbf{0}, +\infty)$ | same as q | Mean absolute relative error** | – |
| MSE | $[\mathbf{0}, +\infty)$ | same as q | Mean square error | Moriasi et al. (2007) |
| RMSE | $[\mathbf{0}, +\infty)$ | same as q | Root mean square error | Barnston (1992); Willmott and Matsuura (2005); Moriasi et al. (2007) |
| NSE | $(-\infty, \mathbf{1}]$ | unitless | Nash-Sutcliffe efficiency | Nash and Sutcliffe (1970) |
| KGE | $(-\infty, \mathbf{1}]$ | unitless | Kling-Gupta efficiency | Gupta et al. (2009) |
| KGE_D | – | unitless | Decomposition of the Kling-Gupta efficiency | Gupta et al. (2009) |
| KGEPRIME | $(-\infty, \mathbf{1}]$ | unitless | Modified Kling-Gupta efficiency | Kling et al. (2012) |
| KGEPRIME_D | – | unitless | Decomposition of the modified Kling-Gupta efficiency | Kling et al. (2012) |
| CONT_TBL | $[0, +\infty)$ | unitless | Contingency table | – |

\* optimal value in bold where applicable

\*\* corresponds to MAE divided by the observed mean

reference is not always clearly defined. Therefore, `evalhyd` only provides the scores for which this is the case: the Nash-Sutcliffe Efficiency (NSE), where the reference is taken as the mean of the observations (i.e. the sample climatology); the Relative Operating Curve Skill Score (ROCSS), where the reference is taken as random forecasts; and the Brier Skill Score (BSS), where the reference is taken as the sample climatology (i.e. constant forecasts of the sample climatological relative frequency, analytically corresponding to the uncertainty term of the Brier Score, see Wilks (2011), eq. 8.43). For all other metrics, the reference needs to be chosen by the user before `evalhyd` can be used to compute the predictions score and the reference score separately. Then, they can be combined following the ratio-based formulation mentioned above to get the skill score. The illustrative example in the following section provides a demonstration of this where the CRPSS is computed using two references provided by the user (the persistence and the climatology). To perform bootstrapping on such custom-made skill scores, the samples must be the same for the predictions score and the reference score, otherwise they will not be compared on the same periods. To do so, the seed used by the bootstrapping functionality can be fixed to the same given value for the computation of both scores.

**Table 3.** Collection of probabilistic metrics available via `evalp` entry point in `evalhyd`.

| Identifier | Range* | Unit | Details | Related references |
|---|---|---|---|---|
| BS | $[\mathbf{0}, +\infty)$ | unitless | Brier score | Brier (1950); Wilks (2011) |
| BSS | $(-\infty, \mathbf{1}]$ | unitless | Brier skill score | Hamill and Juras (2006); Wilks (2011) |
| BS_CRD | $[\mathbf{0}, +\infty)$ | unitless | Calibration-refinement decomposition of the Brier score (i.e. reliability, resolution, and uncertainty) | Wilks (2011) |
| BS_LBD | $[\mathbf{0}, +\infty)$ | unitless | Likelihood-base rate decomposition of the Brier score (i.e. type-2 bias, discrimination, and sharpness) | Wilks (2011) |
| REL_DIAG | NA | unitless | Reliability diagram (i.e. forecast probabilities, observed frequencies, and sampling frequencies) | Weisheimer and Palmer (2014) |
| CRPS_FROM_BS | $[\mathbf{0}, +\infty)$ | same as q | Continuous ranked probability score derived from Brier scores | – |
| CONT_TBL | $[0, +\infty)$ | unitless | Contingency table | – |
| POD | $[0, \mathbf{1}]$ | unitless | Probability of detection | – |
| POFD | $[\mathbf{0}, 1]$ | unitless | Probability of false detection | – |
| FAR | $[\mathbf{0}, 1]$ | unitless | False alarm rate | – |
| CSI | $[0, \mathbf{1}]$ | unitless | Critical success index | – |
| ROCSS | $(-\infty, \mathbf{1}]$ | unitless | Relative operating characteristic skill score | – |
| CRPS_FROM_ECDF | $[\mathbf{0}, +\infty)$ | same as q | Continuous ranked probability score derived from empirical cumulative density function | Hersbach (2000) |
| QS | $[\mathbf{0}, +\infty)$ | same as q | Quantile scores | Gneiting and Raftery (2007) |
| CRPS_FROM_QS | $[\mathbf{0}, +\infty)$ | same as q | Continuous ranked probability score derived from quantile scores | Gneiting and Ranjan (2011) |
| RANK_HIST | NA | unitless | Rank histogram | Talagrand et al. (1997) |
| DS | $[0, +\infty)$ | unitless | Delta score | Candille and Talagrand (2005); Anctil and Ramos (2017) |
| AS | $[0, \mathbf{1}]$ | unitless | Alpha score | Renard et al. (2010) |
| CR | $[0, 1]$ | unitless | Coverage ratio | – |
| AW | $[\mathbf{0}, +\infty)$ | same as q | Average width | – |
| AWN | $[\mathbf{0}, +\infty)$ | | Average width normalised | Bourgin et al. (2015) |
| WS | $[\mathbf{0}, +\infty)$ | same as q | Winkler score | Winkler and Murphy (1979); Gneiting and Raftery (2007) |
| ES | $[\mathbf{0}, +\infty)$ | same as q | Energy score | Gneiting et al. (2008) |

* optimal value in bold where applicable; NA means not applicable

## 6 Illustrative example

In order to illustrate the capabilities of `evalhyd`, open access data sets were chosen from the literature and evaluated. The prediction data chosen are the ones produced by Zsoter et al. (2020) and correspond to river discharge reforecasts from the Global Flood Awareness System (GloFAS) for the period 1999-2018. The observation data chosen are the ones produced by Harrigan et al. (2021) and correspond to river discharge reanalysis data from the GloFAS hydrological modelling chain forced with ERA5 meteorological reanalysis data for the period 1979-2022. In addition, the study by Harrigan et al. (2023) is used as it provides evaluation results for this data set for the period 1999-2018 using the CRPSS computed against two different benchmarks (persistence and climatology). The persistence benchmark is "defined as the single GloFAS-ERA5 daily river discharge of the day preceding the reforecast start date" (Harrigan et al., 2023), while the climatology benchmark is "based on a 40-year climatological sample (1979–2018) of moving 31 d windows of GloFAS-ERA5 river discharge reanalysis values, centred on the date being evaluated" (Harrigan et al., 2023). In this example, the focus is put on the 23 GloFAS stations located in France.

The first objective of this example is to show that the results published by Harrigan et al. (2023) can be reproduced using `evalhyd`. The second objective is to show the many possibilities offered by the functionalities and the metrics of `evalhyd`, which could be used to further analyse the reforecasts, for instance.

### 6.1 Reproducing published results

Figure 5 (Figure A2) provides the GloFAS reforecasts performance against the climatology benchmark (against the persistence benchmark) using the CRPSS. It focuses on four of the 17 lead times available, thus mirroring Figure 6 (Figure 7) in Harrigan et al. (2023). In order to be more precise and exhaustive in the comparison between their published performance and the performance obtained with `evalhyd`, Figure 6 (Figure A3) provides a complete comparison of the performance against the climatology benchmark (against the persistence benchmark) for each station and for each lead time. The data points all lying on the 1:1 line demonstrates that the performance is identical and, therefore, that the published results are indeed successfully reproduced using `evalhyd`.

### 6.2 Showcasing some useful functionalities

The evaluation tool `evalhyd` features many more metrics and functionalities than those used to reproduce the published results above. This section makes use of the same GloFAS reforecasts as an example data set to showcase some additional functionalities available to further explore the performance of streamflow (re)forecasts, e.g. metric uncertainty estimation or focus on certain flow ranges.

#### 6.2.1 Uncertainty analysis using the bootstrapping functionality

In order to estimate the sampling uncertainty on the metric values, `evalhyd` bootstrapping functionality can be used (see subsection 4.5 for details). Figure 7 showcases the results obtained using this bootstrapping functionality with 1000 samples of

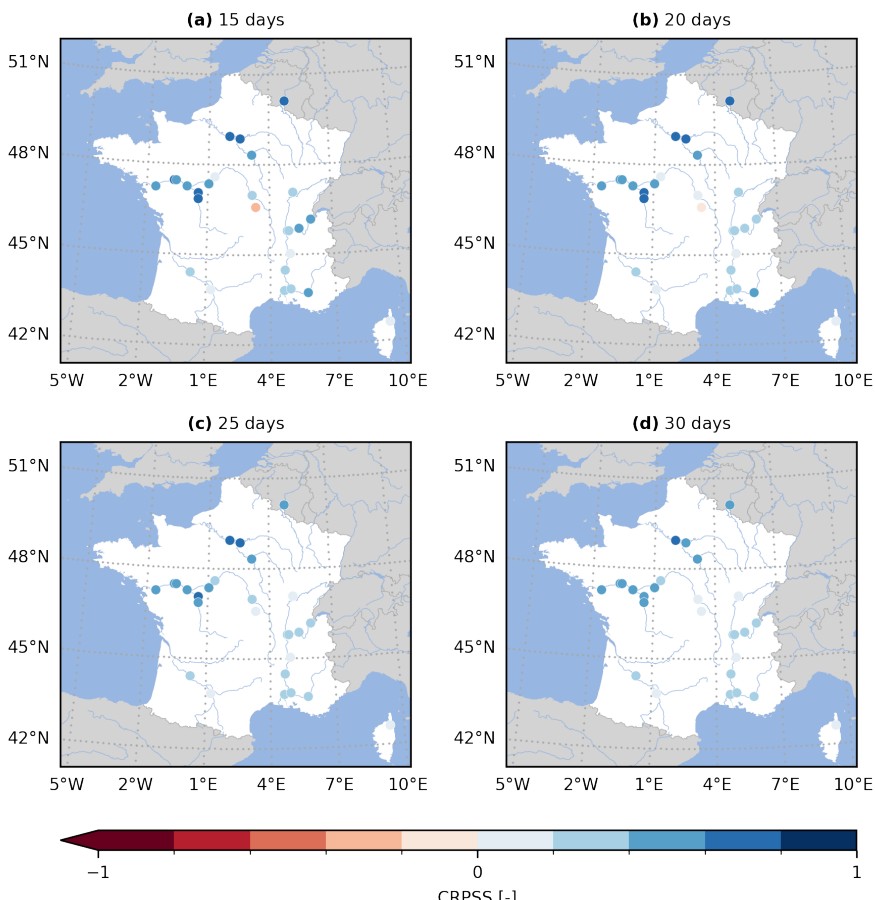

**Figure 5.** Continuous Rank Probability Skill Score (CRPSS) for the GloFAS reforecasts (v2.2) against the climatology benchmark. The CRPSS is computed with `evalhyd` for all 23 GloFAS stations located in France, and for four lead times (i.e. **(a)** 15 d, **(b)** 20 d, **(c)** 25 d, and **(d)** 30 d), mirroring a zoomed-in version of Figure 7 in Harrigan et al. (2023).

10 years each, summarised using a distribution of quantiles and displayed as box-plots. The evaluation metric used is the Brier Skill Score (BSS) against the climatology benchmark to assess the performance of the reforecasts in predicting the exceedance of a threshold set as the 20th observed percentile, and the 12-day lead time is considered.

These results obtained with `evalhyd` can be used to explore the variability in performance across the GloFAS stations by, for example, comparing the median performance (varying from 0.323 to 0.903 here). In addition, the varying widths of the boxes across the GloFAS stations (from 0.030 to 0.139 here) can be used as a measure of the uncertainty associated with the predictive performance of the flood events from one station to another.

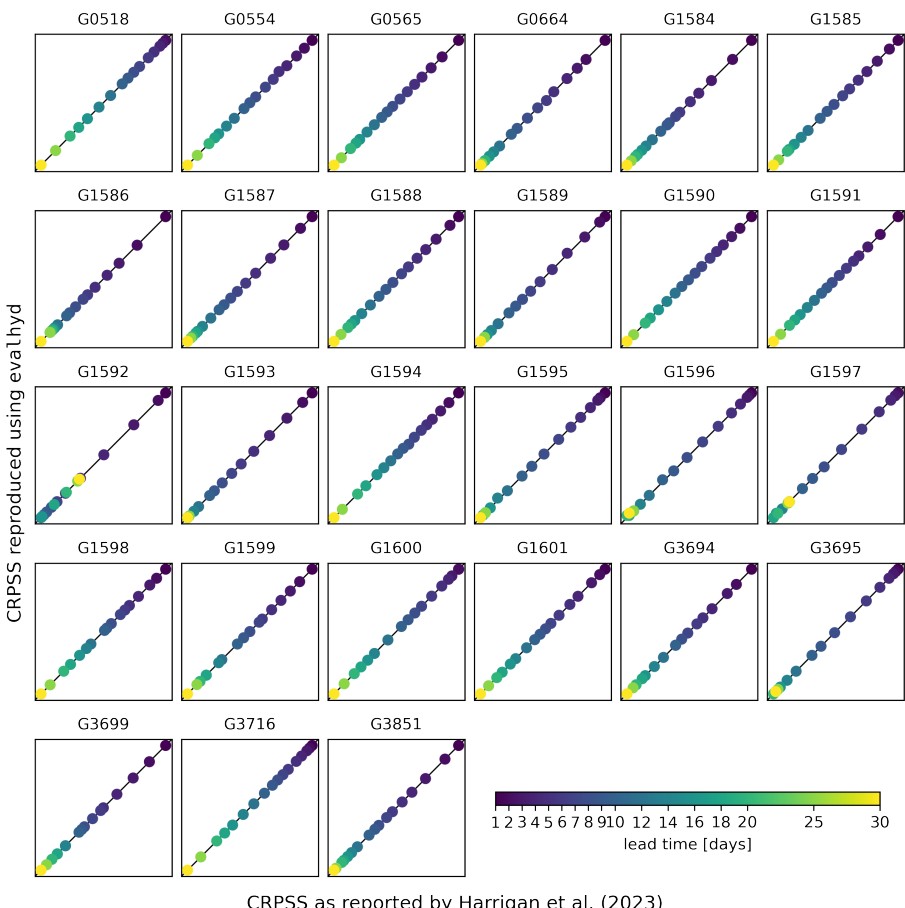

**Figure 6.** Comparison between the CRPSS reported in the Supplement of Harrigan et al. (2023) (x-axis) and the CRPSS computed using `evalhyd` (y-axis) against the climatology benchmark. Each panel represents one of the 23 GloFAS stations located in France, and in each panel, the diagonal represents the 1:1 line, and each data point represents one of the 17 lead times in the GloFAS reforecasts (v2.2).

### 6.2.2   Data stratification using the masking functionality

The predictive performance of streamflow forecasts may vary depending on the flow range considered (e.g. flood forecasting vs. drought forecasting). Bellier et al. (2017) suggest a forecast-based sample stratification for continuous scalar variables in order to consider the merits of streamflow forecasts on different ranges of flows. Such analyses can be easily performed using the conditional masking functionality of `evalhyd`.

Figure 8 provides an example stratifying the rank histogram into three components, one for low-flow periods (using the masking condition "periods where the predicted median is below the 30th predicted percentile"), one for average-flow periods (using the masking condition "periods where the predicted median is between the 30th and the 70th predicted percentiles"),

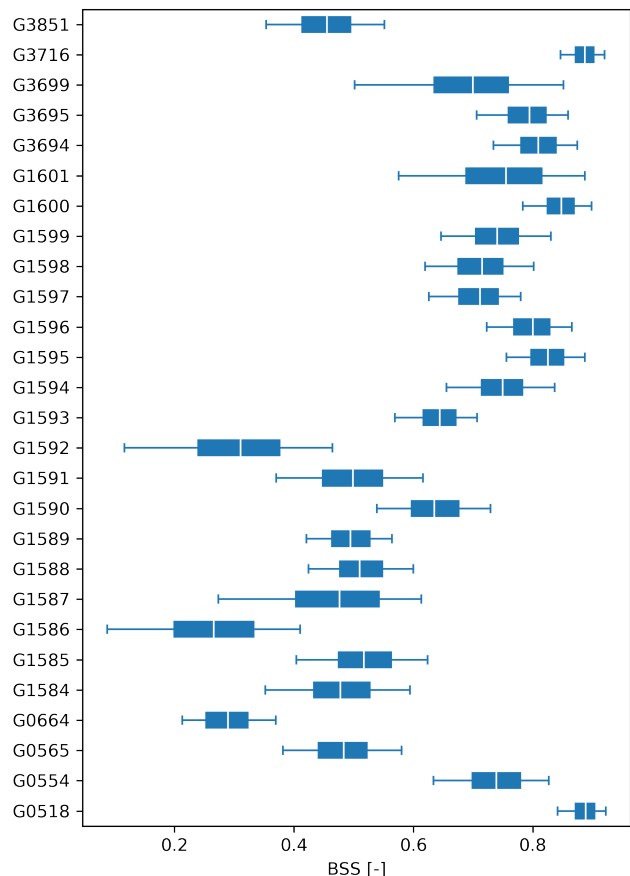

**Figure 7.** Brier Skill Scores (BSS) on the exceedance of a 20th percentile threshold (x-axis) for the GloFAS reforecasts (v2.2) against the climatology benchmark for all 23 GloFAS stations located in France (y-axis) for a 12-day lead time. Each boxplot represents the sampling distribution obtained using the bootstrapping functionality of `evalhyd` (with 1000 samples of 10 years each), where the box is formed of the inter-quartile range (i.e. 25th-75th percentiles) and is split using the median (i.e. 50th percentile), and the whiskers are stretching from the 5th to the 95th percentiles.

and one for high-flow periods (using the masking condition "periods where the predicted median is above the 70th predicted percentile").

These results obtained with `evalhyd` can be used to explore the dispersion of the GloFAS reforecasts. For example, for a given station (G0664, Le Bevinco at Olmeta-di-Tuda) and a given lead time (6-day lead time), the U-shapes of the histograms for low-flow and average-flow conditions suggest an under-dispersion of the reforecasts, while the upslope-shape for high flow conditions suggests a small negative bias.

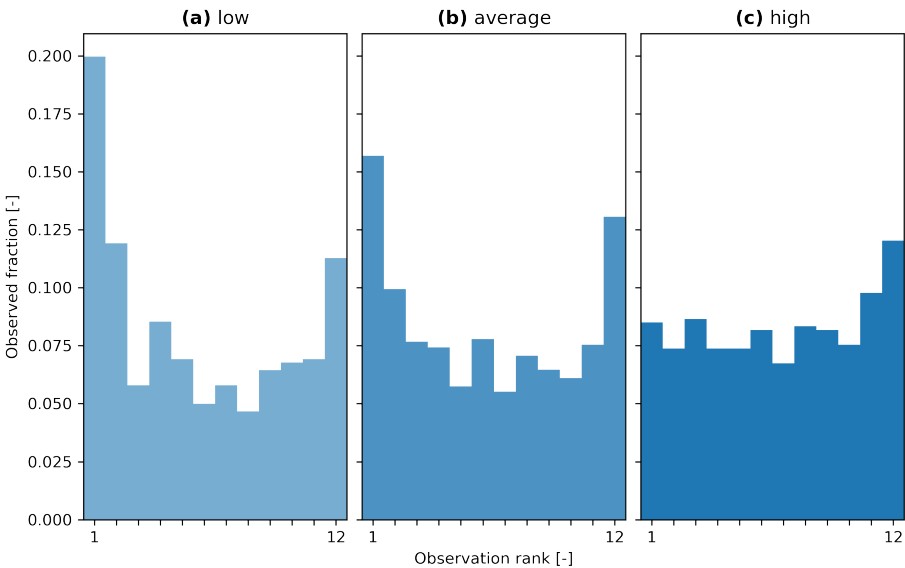

**Figure 8.** Rank histograms for the GloFAS reforecasts (v2.2) for the GloFAS station G0664 (Le Bevinco at Olmeta-di-Tuda) and for the 6-day lead time, stratified using the conditional masking functionality of `evalhyd`: **(a)** for predicted low-flow conditions (i.e. for periods where the predicted median is below the 30th predicted percentile), **(b)** for predicted average-flow conditions (i.e. for periods where the predicted median is above or equal to the 30th predicted percentile, and below or equal to the 70th predicted percentile), and **(c)** for predicted high-flow conditions (i.e. for periods where the predicted median is above the 70th predicted percentile).

### 6.2.3 Multivariate analysis using the multi-dimensional paradigm

Gneiting et al. (2008) proposed the Energy Score (ES) as a multivariate generalisation of the CRPS. This metric makes it possible to aggregate the performance of several study sites in order, for example, to explore regional trends in forecasting performance. The inclusion of multi-variate metrics would not have been possible without the multi-dimensional paradigm chosen for `evalhyd`.

Figure 9 provides a multi-site equivalent of Figure 5 by aggregating the GloFAS stations into six main hydrographic basins in France. The performance is measured against the climatology benchmark using the Energy Skill Score (ESS). Given the varying number of stations in each basin, it is preferred over the Energy Score (ES) to allow for a comparison across basins.

These results obtained with `evalhyd` can be used to explore regional trends. For example, the performance for the four lead times considered suggest that the reforecasts for the Meuse, the Seine, and the Loire river basins are the most skilful, regardless of the lead time considered, while the reforecasts for the Garonne and the Corse river basins are the least skilful.

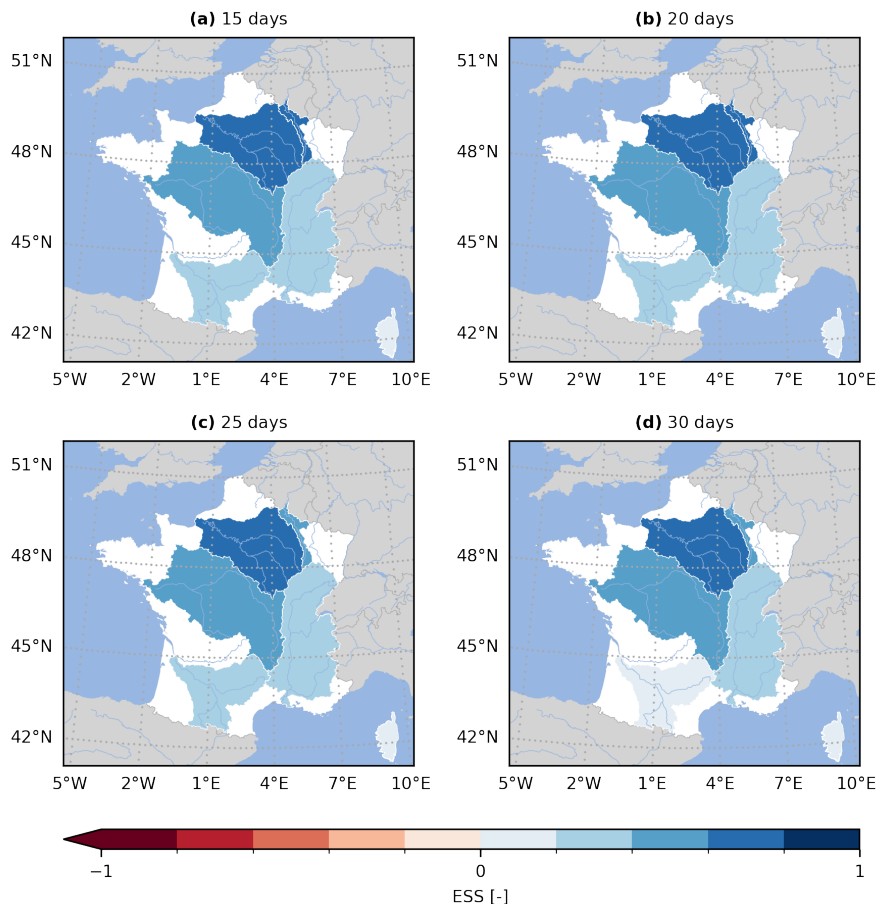

**Figure 9.** Energy Skill Score (ESS) for the GloFAS reforecasts (v2.2) against the persistence benchmark. The ESS is computed using the multi-dimensional design of `evalhyd` for six main French hydrographic basins (from North to South, Meuse, Seine, Loire, Rhône, Garonne, Corse), and for four lead times (i.e. **(a)** 15 d, **(b)** 20 d, **(c)** 25 d, and **(d)** 30 d).

## 7  Limitations of the tool

Some limitations in the current version of `evalhyd` exist. In a hydro-meteorological context, streamflow forecasts are often produced as deterministic forecasts or as ensemble forecasts to estimate the predictive probability distributions. However, they can also be issued as continuous predictive probability distributions. `evalhyd` only offers a solution for the first two situations, and the third situation is not currently supported.

As part of the design process, there was a focus on computational efficiency. This lead to the decision to rely on a compiled language and to resort to memoisation. Arguably, the former complicates the development effort compared with interpreted languages. In addition, the latter complicates the algorithms because the metric computations need to be decomposed, and the temporal reduction needs to be delayed for the masking functionality. Together, these design decisions hamper contributions

from the hydrological community, e.g. the inclusion of additional metrics. We believe that this is an unavoidable compromise for the sake of efficiency. Nevertheless, beyond efficiency considerations, relying on a compiled language also offers easier and cleaner options for exposing the metrics and the functionalities to several interpreted languages, instead of calling one interpreted language from another, for instance.

Furthermore, another design decision was to focus on the numerical aspect of streamflow evaluation, leaving aside its visualisation aspect (e.g. plotting rank histograms, reliability diagrams, and so on), unlike existing tools such as EVS, which offers a graphical user interface (Brown et al., 2010). Beyond a healthy separation of concerns, this is also partly influenced by the fact that the compiled core is intended to feature all of the functionalities presented here, and that visualisation capabilities are more accessible in interpreted languages. Nonetheless, the data necessary to plot such figures can be provided as numerical values to limit the effort on the user's side. This is already the case in `evalhyd` with the rank histogram (using the metric `RANK_HIST`) and the reliability diagram (using the metric `REL_DIAG`).

Finally, the conditional masking functionality currently available in `evalhyd` only makes it possible to perform unilateral conditional evaluation, that is to say that conditions can only be applied to the observations or to the predictions, but not to both at the same time (i.e. bilateral conditional evaluation). Unilateral conditioning may lead to synthetic bias in the evaluation. For instance, if the condition is applied to predicted values exceeding a given flood threshold, both "hits" and "false alarms" (in a contingency table sense) will be considered, whereas bilateral conditioning makes it possible to only consider the "hits" (Casati, 2023). In addition, the conditional masking functionality is only applicable to the variable being evaluated (i.e. streamflow), and not to an independent variable. For instance, one may want to evaluate streamflow predictions only for days with exceptionally intense rainfall, which is not possible with the conditional masking functionality as it stands. This may lead to further synthetic bias when considering conditions on extreme predicted values, which are bound to include both extreme observed values, but also more average ones, artificially accentuating the over-predictive character of the forecasts (Casati, 2023). Nonetheless, these two limitations with the conditional masking functionality can actually be avoided by favouring the temporal masking functionality, where the user is free to create their own masks using the conditions of their choice.

# 8 Conclusions and perspectives

In this article, a new evaluation tool for streamflow predictions, named `evalhyd`, is presented. The current version of this tool gives hydrologists access to a large variety of evaluation metrics, commonly used to analyse streamflow predictions. It also offers convenient and hydrologically-relevant functionalities such as sample stratification or metric uncertainty estimation. The tool is readily available to a diversity of users, as it is distributed as a header-only C++ library, as a Python package, as an R package, and as a command line tool. These packages are all available on conda-forge (https://conda-forge.org, last access: 30 Jan 2024), the Python package is also available from PyPI (https://pypi.org/project/evalhyd-python, last access: 30 Jan 2024), and the R package is also available from R-universe (https://hydrogr.r-universe.dev/evalhyd, lass access: 30 Jan 2024). These packages come with extensive online documentation, accessible at https://hydrogr.github.io/evalhyd (last access: 30 Jan 2024).

The main limitations identified for the tool are the lack of a visualisation functionality, the lack of support for the evaluation of continuous probability distributions, and the limited scope for extensibility by non-expert programmers.

Some of the developments envisaged for future versions of the tool include the addition of other evaluation metrics, especially multivariate ones, the implementation of additional bindings for other open source languages (e.g. Julia, Octave), the addition of other preliminary processing options (e.g. computation of commonly considered high-flow and low-flow statistics, summary statistics on sliding windows, etc.) and the support of configuration files common across the software stack to further simplify collaboration and reproducibility.

Through its polyglot character, this tool is aimed at the hydrological community as a whole, and we hope that it can foster collaborations amongst its users without a programming language barrier. The organisation of user workshops could lead to new evaluation metrics and new evaluation strategies, for example based on bilateral conditioning, that could then be implemented in the tool to directly benefit our community at large.

*Code availability.* The package used for the illustrative example is `evalhyd-python`. It is available from HAL (https://hal.science/hal-04088473, last access: 30 Jan 2024) (Hallouin and Bourgin, 2024a). `evalhyd` user guide and tutorials are available in the HTML documentation archived on Software Heritage (Hallouin and Bourgin, 2024b). The scripts used to produce the figures in the illustrative example are available on Zenodo (Hallouin, 2024).

*Data availability.* The observation data used in this study, i.e. the GloFAS-ERA5 v2.1 river discharge reanalysis data, can be downloaded from Copernicus' Climate Data Store (https://doi.org/10.24381/cds.a4fdd6b9, last access: 10 May 2023) (Harrigan et al., 2021). The prediction data used in this study, i.e. the GloFAS v2.2 river discharge reforecast data, can also be downloaded from Copernicus' Climate Data Store (https://doi.org/10.24381/cds.2d78664e, last access: 10 May 2023) (Zsoter et al., 2020).

*Author contributions.* CP, FB, MHR, and VA were responsible for the funding acquisition. All co-authors contributed to the conceptualisation. TH and FB developed the software. TH performed the data curation and the formal analysis. TH prepared the original draft of the manuscript. All co-authors contributed to the review and editing of the manuscript.

*Competing interests.* The authors declare that none of them has any competing interests.

*Acknowledgements.* The authors acknowledge the financial support of the French Ministry for the Environment (DGPR/SNRH/SCHAPI) for this work. The authors would like to thank Shaun Harrigan from ECMWF for the help with pre-processing the GloFAS reforecasts,

Antoine Prouvost and Johan Mabille from QuantStack for the help with using the `xtensor` software stack, and Guillaume Thirel, Laurent Strohmenger, and Léonard Santos from INRAE for their feedback on this paper.

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

**(a)** C++ interface

```
#include <xtensor/xtensor.hpp>
#include <evalhyd/evalp.hpp>
xt::xtensor<double, 2> obs =
{{4.7, 4.3, 5.5, 2.7, 4.1}};
xt::xtensor<double, 4> prd =
{{{{5.3, 4.2, 5.7, 2.3, 3.1},
{4.3, 4.2, 4.7, 4.3, 3.3},
{5.3, 5.2, 5.7, 2.3, 3.9}}}};
xt::xtensor<double, 2> thr = {{4., 5.}};

auto res = (
evalhyd::evalp(obs, prd, {"BS"}, thr, "high")
);
```

**(b)** Python interface

```
import numpy
import evalhyd
obs = numpy.array([[4.7, 4.3, 5.5, 2.7, 4.1]])
prd = numpy.array([[[[5.3, 4.2, 5.7, 2.3, 3.1],
[4.3, 4.2, 4.7, 4.3, 3.3],
[5.3, 5.2, 5.7, 2.3, 3.9]]]])
thr = numpy.array([[4., 5.]])

res = (
evalhyd.evalp(obs, prd, ["BS"], thr, "high")
)
```

**(c)** R interface

```
library(evalhyd)
obs ← rbind(c(4.7, 4.3, 5.5, 2.7, 4.1))
prd ← array(
rbind(c(5.3, 4.2, 5.7, 2.3, 3.1),
c(4.3, 4.2, 4.7, 4.3, 3.3),
c(5.3, 5.2, 5.7, 2.3, 3.9)),
dim=c(1, 1, 3, 5)
)
thr ← rbind(c(4., 5.))

res ← (
evalhyd::evalp(obs, prd, c("BS"), thr, "high")
)
```

**(d)** Command line interface

```
cat "./obs/site_a.csv"
    4.7,4.3,5.5,2.7,4.1
cat "./prd/leadtime_1/site_a.csv"
    5.3,4.2,5.7,2.3,3.1
    4.3,4.2,4.7,4.3,3.3
    5.3,5.2,5.7,2.3,3.9
cat "./thr/site_a.csv"
    4.,5.

res=$(evalhyd evalp \
"./obs/" "./prd/" "BS" \
--q_thr "./thr/" --events "high")
```

**Figure A1.** Comparison of the interfaces for the probabilistic entry point `evalp` across the `evalhyd` software stack through a simple example evaluating ensemble predictions (`prd`, of shape $\{sites : 1,\ lead\ times : 1,\ ensemble\ members : 3,\ time : 5\}$) against observations (`obs`, of shape $\{sites : 1,\ time : 5\}$) using the Brier score (BS) based on streamflow thresholds (`thr`, of shape $\{sites : 1,\ thresholds : 2\}$) for flood events (i.e. high flows): **(a)** C++ interface fed with `xtensor` data structures, **(b)** Python interface fed with `numpy` data structures, **(c)** R interface fed with R data structures, and **(d)** command line interface fed with data in CSV files in structured directories.

Yuan, X. and Wood, E. F.: On the clustering of climate models in ensemble seasonal forecasting, Geophysical Research Letters, 39, https://doi.org/10.1029/2012GL052735, 2012.

Zsoter, E., Harrigan, S., Barnard, C., Blick, M., Ferrario, I., Wetterhall, F., and Prudhomme, C.: Reforecasts of river discharge and related data by the Global Flood Awareness System. v2.2, https://doi.org/10.24381/cds.2d78664e, 2020.

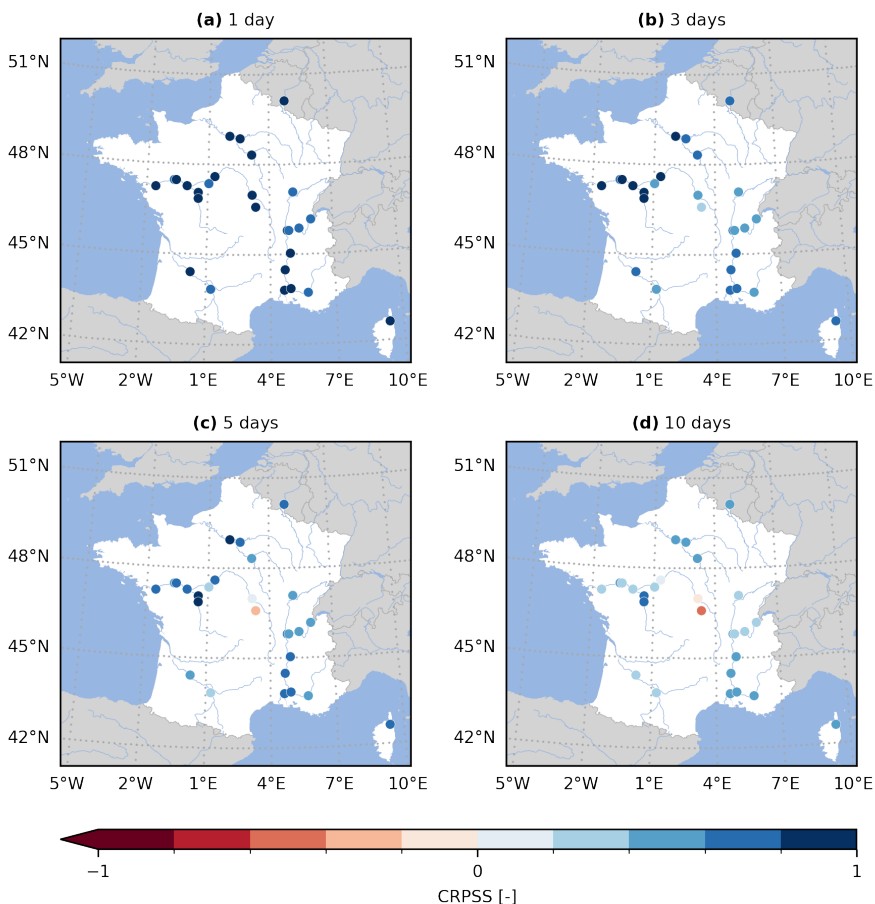

**Figure A2.** Continuous Rank Probability Skill Score (CRPSS) for the GloFAS reforecasts (v2.2) against the persistence benchmark. The CRPSS is computed with `evalhyd` for all 23 GloFAS stations located in France, and for four lead times (i.e. **(a)** 1 d, **(b)** 3 d, **(c)** 5 d, and **(d)** 10 d), mirroring a zoomed-in version of Figure 6 in Harrigan et al. (2023).

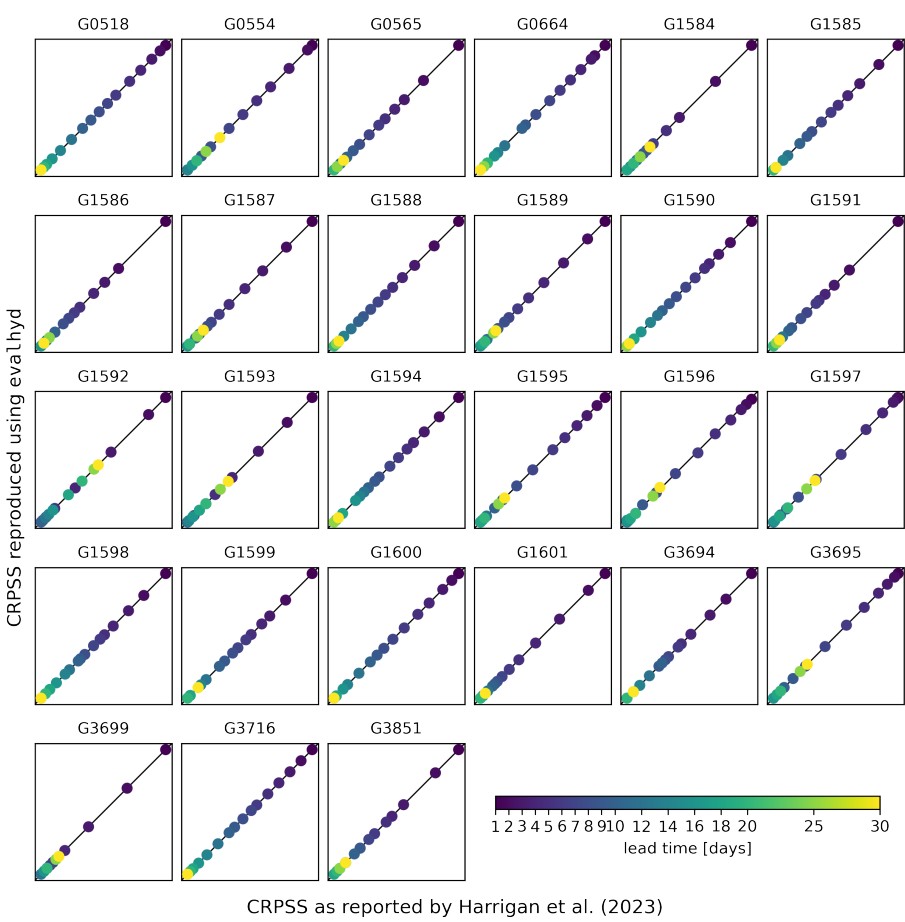

**Figure A3.** Comparison between the CRPSS reported in Harrigan et al. (2023) (x-axis) and the CRPSS computed using `evalhyd` (y-axis) against the persistence benchmark. Each panel represents one of the 23 GloFAS stations located in France, and in each panel, the diagonal represents the 1:1 line, and each data point represents one of the 17 lead times in the GloFAS reforecasts (v2.2).