# Peer review of "evallyd v0.1.2: a polyglot tool for the evaluation of deterministic and probabilistic streamflow predictions"

_EGUsphere, 2023_

## Referee Comment (RC1)

**Revisions for the manuscript "evalhyd v0.1.1: a polyglot tool for the evaluation of deterministicand probabilistic streamflow prediction" by Hallouin et al.**

**General comments**

This article presents a hydrological verification tool which is coded in several languages to accommodate different users used to different coding habits. The tool includes some data pre-processing capabilities, as well as inference by bootstrapping and the possibility of applying conditional verification (e.g. tailoring different streamflow regimes). The article reads very well and is logically structured; the language, figures and text are very clear; the results shown corroborate the conclusions.

I really enjoyed reading this article! and I have no hesitation in recommending it for publication. Here below I outline some minor suggestions, for the author's considerations, after which the manuscript can be published.

**Specific comments**

1. Line 105, the dimensions for a deterministic evaluation *{series,time}* seems to me inconsistent with the dimensions for the probabilistic evaluation *{sites, leadtimes, ensemble members, time}*, introduced at line 99. I would add "*sites, leadtimes*" also for the deterministic evaluation to have *{sites, leadtimes, series, time}*. (essentially you have replaced the "*series*" of multiple simulations to "*ensemble members*")

2. I have few questions about the bootstrapping: at lines 166-167 I understand that each year is a block, but I do not understand what are the sub-periods: can you please explain. Do you perform a resampling with replacement? (allowing a year to be selected multiple times in the same bootstrap sample). At lines 183-184 you mention that the user can evaluate a custom-made skill score with the reference of his/her choice (and that evalhyd evaluates the two scores separately, prior evaluating the skill score): do you use a pair-bootstrapping for the confidence intervals on this skill score, right? (aka the bootstrap re-samples are the same for the prediction and the reference).

3. In the example showcased in section 6 you compare the prediction against persistence. Can you please specify what is the persistence forecast: is it a fix persistence (e.g. for each validity date you consider as prediction the streamflow of n days before, with n fix), or does the persistence align with the prediction lead-time, so that a 10 day forecast is compared to a 10 day persistence, and a 2 day forecast is compared against a 2 day persistence? (essentially the prediction is compared to its initial state). Thank you for specifying this in the article.

4. Figure 4 and A3: the CRPSS against climatology consistently decreases with lead-time (as expected), for almost all stations, whereas the CRPSS against persistence does not: why? Is there a sort of re-emergence of the signal which renders the persistence more skillful for some time-lags?

5. Table 2: please define the Mean Absolute Relative Error (or provide a reference).

6. Why for the deterministic prediction the contingency table entries are provided (Table 2 last line), whereas for the probabilistic prediction (Table 3) you compute directly POD, POFD, FAR, CSI? Would not be more symmetric to provide also for the probabilistic metrics the contingency table entries?

7. **Conditional masking: in light of the comments in the main discussion item on conditional verification, please consider adding some text in the article about the effects of bilateral versus unilateral conditions when performing conditional verification.**

8. Figure 7 shows regional scores for the river basins: The Loire basin seems performing very well, however in Fig.3 (and A2) if was shown that for two of the upstream stations the skill was quite poor. Maybe we should not aggregate regionally? I suggest to remove this option and section 6.2.3.

**Technical corrections**

1. Especially at the beginning of the article, it is not very clear what is meant with "post-processing of the computed metrics" (it becomes clear only after reading Section 6). Since the term "post-processing" is usually used for statistically correct a forecast, I suggest using different phrasing.
    a. Line 3 can be rephrased as " … , but it can also involve the data quality control and pre-processing, as well as performing further analysis (e.g. tailored stratification, inference) on the computed metrics".
    b. Line 29 can be rephrased as " … , the metrics can be subject also to sensitivity analysis and uncertainty estimation"

2. In section 3.1 (e.g. lines 72, 80, 83) you talk about "satellite bindings": can you please avoid using "satellite", and maybe just refer to these as "bindings" or (even better) "extensions" (as they are named `xtensor-python, xtensor-r,` etc.)

3. Line 111-112: the "the quadratic error between the observations and their arithmetic mean" is the observation variance, right? I suggest mentioning this, e.g. in a parenthesis following the text.

4. Lines 123-128 need rephrasing, I'll try rephrase some text (as I understood the issue), e.g. line 124: "In addition, when predictions for several lead-times are considered, the predictions with validity time beyond the observed timeseries need to be flagged as *not a number*." At line 126 I suggest not using "invalid dates" but rather write " … and dates where the observation has no matched prediction must be identified as *not a number*". I also suggest referring to *initialization + lead time = validity time* for the prediction to be matched with the observation at the validity time.

5. The caption of Fig.2 need rephrasing too, in view of the previous comment. Moreover, it is the last row which shows the observation (and not the first), and the prediction is on the preceding (and not following) rows. What are the numbers within the rounded rectangles?

6. Lines 180 and 182: the reference for the NSE is indeed the mean of the observations (aka the sample climatology). The reference for the BSS is (correctly) again the sample climatology. However, from the way these sentences are phrased, these two references seem different references. You might want to rephrase your text to fix this, e.g. line 180 could be " … where the references is taken as the mean of the observations (aka the sample climatology)"
7. Caption of figure 6: please specify that the condition is applied only for the forecast, e.g. you can write " (a) for predicted low-flow conditions, … ".
8. Line 268: use the present tense "is presented" (and not "was presented").

---

## Author Response (AR1)

Dear Editor, Dear Referees,

We would like to thank you for considering our manuscript, for the thorough review, and for the constructive feedback you provided. Please find below a detailed point-by-point answer to the referees' comments.

The authors.

**Nomenclature:**

RXCY – Referee number X Comment number Y
AR – Authors' Reply
O-LX – Original manuscript Line X
R-LX – Revised manuscript Line X

**Referee #1 (Barbara Casati)**

**General comments**

**R1C1:** I really enjoyed reading this article, and *evalhyd* seems a very nice verification too: felicitations! I have uploaded a file with some minor suggestions, while here with this following comment I wish to bring the attention of the scientific community to one aspect of verification, for the general online discussion.

I was particularly triggered about the option in *evalhyd* of performing conditional verification. I will share here some of our own experience (at the Canadian Met Service) with conditional verification, which maybe can inspire further developments in the tool and, more in general, awareness in the interpretation of the results.

Conditioning on the verification sample can have strong impacts on the verification results (e.g. it can flip the sign of a bias), and hence allows in-depth analysis and understanding of the prediction performance, since the conditioning is usually related to physically-driven phenomena. In a sense, conditional verification is the first step towards process-based diagnostics.

In verification exercises which include several variables (e.g. pressure, temperature, clouds, etc) applying a condition to a variable while verifying a different variable is the common practice (as an example, verification of surface temperature in cloudy versus clear-sky conditions inform of the model performance in reproducing the radiation budget). The condition, however, should be applied to both observed and forecast values (e.g. forecast AND observation being cloudy): I will refer to this double condition as *bilateral*. On the other hand, when a *unilateral* condition is applied, to only the observed or forecast variable (e.g. cloudy conditions only for the forecast) this can synthetically introduce a bias in the verification results: in the cloud/temperature example above, stratifying for cloudy conditions only in the forecast leads to a synthetic warm bias for the surface temperature, because in the sample there are bound to be both cloudy

and clear sky observations, and when the observations have clear sky the surface temperature is expected to be colder. In other words, the bilateral condition will sample all the "hits" for cloudy sky, whereas the unilateral condition will sample the "hits" and "false alarms" for cloudy sky. From our experience, we advise bilateral over unilateral conditioning. (Of course one can also do unilateral conditioning, but need to be aware of the introduced biases in the interpretation of the verification results).

Applying the unilateral condition to the same variable which is verified might also lead to synthetic biases. As an example, if you stratify your sample for the strong *predicted* stream flows, you are bound to include in the sample several strong observed stream flows (the "hits"), but also some average or weak observed stream-flows (because the prediction might have some "false alarms"). Then you tend to "artificially" diagnose over-prediction for the strong stream flow (and vice-versa for the low stream flow, conditioning only on the prediction you are bound to find under-estimation, because in your sample you'll have some observed events which are medium or strong).

(...)

I would be grateful if you could add in the article some discussion about unilateral versus bilateral conditions.

**AR:** Thank you for sharing your experience on conditional verification and for the very detailed explanations on the potential biases that unilateral conditioning can lead to. We have added some references in the introductive paragraph of the masking functionality section (see R-L137-140) to give some context on the motivation behind conditional verification. In addition, we have added a paragraph in the limitations of the tool mentioning the risk of introducing synthetic bias when performing unilateral masking and/or applying the condition on the variable being evaluated (see R-L290-301), and we have added a cross-reference to the limitations section in the masking functionality section for the reader not to miss them. We hope that these additions are faithful to your explanations.

**R1C2:** I was amazed (also a bit puzzled) to see that in your Figure 6 you have opposite results than I expected (underprediction for high predicted stream flow, more overprediction for low predicted stream flows; the under-dispersion for the average predicted stream flow is instead expected). For me it would be interesting to understand why, is it due to the characteristics of streamflow prediction (where the timing is always predicted well, and hence false alarms and

misses are very rare)? what is the behaviour in the other stations? What would you obtain with the bilateral condition?

**AR:** This figure seems relatively coherent with what is expected given the difficulties in capturing the hydrological variability: models tend to overestimate low flows and to underestimate high flows. In order to provide in-depth explanations for these results, it would be necessary to also consider rainfall predictions to be able to distinguish the quality of the model from the quality of the predictions. This goes beyond the scope of this technical article that only intends to demonstrate what can be produced with the tool, and not to analyse the reforecasts used as an example data set that is well-known and openly accessible.

**Specific comments**

**R1C3:** Line 105, the dimensions for a deterministic evaluation {series,time} seems to me inconsistent with the dimensions for the probabilistic evaluation {sites, leadtimes, ensemble members, time}, introduced at line 99. I would add "sites, leadtimes" also for the deterministic evaluation to have {sites, leadtimes, series, time}. (essentially you have replaced the "series" of multiple simulations to "ensemble members")

**AR:** As mentioned in section 3.3, the four-dimensional character of the inputs for the probabilistic evaluation is motivated by the existence of multi-variate probabilistic metrics (e.g. the energy score proposed by Gneiting et al. (2008)). This is not commonly the case for deterministic evaluation, as there is by definition only one forecast member, and there is no multi-site metric in common use to the best of our knowledge. This is why we decided to keep the dimensionality of the inputs as low as possible. Indeed, deterministic metrics are often used in optimisation problems, where one-dimensional inputs would be sufficient (to compare one simulation series against one observation series), so it appeared to us that it would be heavy for such case to include so many extra dimensions, potentially confusing users only interested in determistic evaluation and not familiar with ensemble forecasting concepts. The "series" dimension can perfectly be used to provide deterministic forecasts for several lead times. But, as mentioned above, the use of the term "leadtimes" for the deterministic evaluation would be reductive to a sole forecast context. So we believe that retaining the dimensionalities as they currently are is the best compromise. We have added forecasting mentions in this section for completeness and clarity (see R-L110-113)

**R1C4:** I have few questions about the bootstrapping: at lines 166-167 I understand that each year is a block, but I do not understand what are the sub-periods: can you please explain. Do

you perform a resampling with replacement? (allowing a year to be selected multiple times in the same bootstrap sample).

**AR:** The sub-periods are made of randomly drawn year blocks. In the revised manuscript, we have reformulated our explanations in the text (see R-L178-181) and added a figure to illustrate the bootstrapping functionality (see Figure 4 in the revised manuscript). Yes, the tool does perform a sampling with replacement. We followed the recommendations made by Clark et al. (2021) to use a non-overlapping block boostrapping, i.e. to draw complete years with replacement. We have added this information to the manuscript (see R-L179-180).

**R1C5:** At lines 183-184 you mention that the user can evaluate a custom-made skill score with the reference of his/her choice (and that evalhyd evaluates the two scores separately, prior evaluating the skill score): do you use a pair-bootstrapping for the confidence intervals on this skill score, right? (aka the bootstrap re-samples are the same for the prediction and the reference).

**AR:** Regarding custom-made skill scores, you are absolutely right in wondering whether the sampling is consistent between the computation of both scores before combining them, as we believe it should be. Thank you for raising this. The tool does allow to keep the same sampling by setting the seed of the pseudo-number generation used for the random sampling with replacement. So, while this cannot be enforced by the tool, this is possible (and essential) to set the same seed when computing both metrics. We have added this information and this recommendation to the manuscript (see R-L203-206).

**R1C6:** In the example showcased in section 6 you compare the prediction against persistence. Can you please specify what is the persistence forecast: is it a fix persistence (e.g. for each validity date you consider as prediction the streamflow of n days before, with n fix), or does the persistence align with the prediction lead-time, so that a 10 day forecast is compared to a 10 day persistence, and a 2 day forecast is compared against a 2 day persistence? (essentially the prediction is compared to its initial state). Thank you for specifying this in the article.

**AR:** The persistence forecast benchmark corresponds to the latter, i.e. the prediction is compared to the same initial conditions for all lead times. We have specified this in the article by directly quoting Harrigan et al. (2023) and, for completeness, we have also added details about the climatology benchmark (see R-L214-218).

**R1C7:** Figure 4 and A3: the CRPSS against climatology consistently decreases with lead-time (as expected), for almost all stations, whereas the CRPSS against persistence does not: why? Is there a sort of re-emergence of the signal which renders the persistence more skillful for some time-lags?

**AR:** This behaviour is discussed in detailed in the peer-review discussion of Harrigan et al. (2023), it is accessible at https://hess.copernicus.org/preprints/hess-2020-532/hess-2020-532-AC1-supplement.pdf (last access: 21 Jan 2024, see discussion for the second specific comment). In short, this is due to the fact that, while both the reforecast and the persistence benchmark deteriote with increasing lead times, the decline in the persistence accuracy happens at a much faster rate than the accuracy of the reforecast. Again, we believe that these aspects are important but they go beyond the scope of a technical paper such as ours.

**R1C8:** Table 2: please define the Mean Absolute Relative Error (or provide a reference).

**AR:** This corresponds to the Mean Absolute Error (MAE) divided by the observed mean, we have added this information as a table footnote (see Table 2 in the revised manuscipt)..

**R1C9:** Why for the deterministic prediction the contingency table entries are provided (Table 2 last line), whereas for the probabilistic prediction (Table 3) you compute directly POD, POFD, FAR, CSI? Would not be more symmetric to provide also for the probabilistic metrics the contingency table entries?

**AR:** Yes, we agree with that. We have added the contingency table as a metric (CONT_TBL) in the probabilistic entry point of evalhyd (available in v0.1.2). Table 3 in the revised manuscript now features CONT_TBL as an available probabilistic metric. Thank you for the suggestion.

**R1C10:** Conditional masking: in light of the comments in the main discussion item on conditional verification, please consider adding some text in the article about the effects of bilateral versus unilateral conditions when performing conditional verification.

**AR:** We have addressed this comment together with the first comment, please refer to our reply for **R1C1**.

**R1C11:** Figure 7 shows regional scores for the river basins: The Loire basin seems performing very well, however in Fig.3 (and A2) if was shown that for two of the upstream stations the skill was quite poor. Maybe we should not aggregate regionally? I suggest to remove

**AR:** We agree that multi-variate scores such as the energy score can hide poorly predicted stations, however, this is the drawback with any aggregation approach, as is the case when interpreting results for large sample hydrology for instance. While such metric should be employed with care, we do believe that it can have some relevance in some applications. In particular, unlike a straightforward mean of CRPS values across stations, the Energy Score employs a weighted Euclidian distance between the different stations considered. We verified our results here and decided to maintain this section for these reasons.

**Technical corrections**

**R1C12:** Especially at the beginning of the article, it is not very clear what is meant with "post-processing of the computed metrics" (it becomes clear only after reading Section 6). Since the term "post-processing" is usually used for statistically correct a forecast, I suggest using different phrasing.

Line 3 can be rephrased as " ... , but it can also involve the data quality control and pre-processing, as well as performing further analysis (e.g. tailored stratification, inference) on the computed metrics".

Line 29 can be rephrased as " ... , the metrics can be subject also to sensitivity analysis and uncertainty estimation"

**AR:** Thank you for highlighting this potential confusion due to our choice of words. Throughout the manuscript we have replaced "pre-processing" by "prelimary processing" and "post-processing" by "subsequent processing", while providing early on in the introduction examples of each type of processing (see R-L22-24).

**R1C13:** In section 3.1 (e.g. lines 72, 80, 83) you talk about "satellite bindings": can you please avoid using "satellite", and maybe just refer to these as "bindings" or (even better) "extensions" (as they are named xtensor-python, xtensor-r, etc.)

**AR:** Given the geoscientific scope of the article, we understand that the use of this particular word figuratively may have been unwise. We followed your piece of advice and dropped satellite in favour of simply "bindings" (see R-L73-90). Since "extension" implies the addition of functionality, which is not the case here, we preferred not to use this term.

**R1C14:** Line 111-112: the "the quadratic error between the observations and their arithmetic mean" is the observation variance, right? I suggest mentioning this, e.g. in a parenthesis following the text.

**AR:** That is right. We have simplified by replacing it directly by the "observed variance" (see R-L118), thank you for the suggestion.

**R1C15:** Lines 123-128 need rephrasing, I'll try rephrase some text (as I understood the issue), e.g. line 124: "In addition, when predictions for several lead-times are considered, the predictions with validity time beyond the observed timeseries need to be flagged as not a number." At line 126 I suggest not using "invalid dates" but rather write " ... and dates where the observation has no matched prediction must be identified as not a number". I also suggest referring to initialization + lead time = validity time for the prediction to be matched with the observation at the validity time.

**AR:** This is indeed a part we found tricky to formulate. Thank you for your suggestions. The term "valid date" was inspired by the CF conventions (https://cfconventions.org/Data/cf-conventions/cf-conventions-1.11/cf-conventions.html#scalar-coordinate-variables, last access: 16-12-2023): "Multiple forecasts from a single analysis (...) The analysis time is identified by the standard name "forecast_reference_time" while the valid time of the forecast is identified by the standard name 'time'." In fact, "valid_time" is the field used by ECMWF in their forecast. However, for clarity, we propose the following rephrasing: "Therefore, when several lead times are considered at once, a temporal shift of the predictions must be applied, and observed dates for which a forecast is not made (i.e. where date ≠ forecast issue date + lead time) must be identified as *not a number*." (see R-L131-133).

**R1C16:** The caption of Fig.2 need rephrasing too, in view of the previous comment. Moreover, it is the last row which shows the observation (and not the first), and the prediction is on the preceding (and not following) rows. What are the numbers within the rounded Rectangles?

**AR:** The caption was rephrased accordingly, as per our reply to **R1C15**: "before and/or after the prediction series to align the prediction validity dates (i.e. issue date + lead time) with the observation dates when several lead times are considered at once". We have also fixed our mistake with the relative locations of the predictions and the observations (the former is indeed below the latter). Thank you for spotting this.

**R1C17:** Lines 180 and 182: the reference for the NSE is indeed the mean of the observations (aka the sample climatology). The reference for the BSS is (correctly) again the sample climatology. However, from the way these sentences are phrased, these two references seem different references. You might want to rephrase your text to fix this, e.g. line 180 could be " … where the references is taken as the mean of the observations (aka the sample climatology)"

**AR:** Yes, this is worth using the same terminology between those two references indeed. As suggested, we have added in parentheses for the NSE that its reference is the sample climatology (see R-L196).

**R1C18:** Caption of figure 6: please specify that the condition is applied only for the forecast, e.g. you can write " (a) for predicted low-flow conditions, … ".

**AR:** Yes, as we performed unilateral conditioning, we have prefixed with "predicted" for sub-labels (a), (b), and (c) of Figure 6 in the revised manuscript.

**R1C19:** Line 268: use the present tense "is presented" (and not "was presented")

**AR:** This is done (see R-L303).

**Referee #2 (Anonymous)**

**General comments**

**R2C1:** The paper introduces evalhyd, an interesting software tool designed for the evaluation of streamflow predictions. The tool's commitment to standardization and open-source accessibility is commendable, providing a valuable contribution to enhancing reproducibility in hydrological studies. Notably, the well-thought-out design principles, which incorporate a compiled C++ core and thin bindings for multiple languages, contribute to the tool's efficiency and usability.

However, despite the paper positioning evalhyd as a contribution to hydroinformatics, the manuscript's focus on the technical aspects of model development limits its scientific impact. The paper could benefit from a more explicit emphasis on the broader scientific implications and advancements in hydrologic science that the tool facilitates.

(…)

Overall more critical analysis is needed on how evalhyd improves on the current state of the art in hydrologic evaluation tools. The paper currently lacks motivation and innovation.

**AR:** We thank the reviewer for the comments and suggestions. The tool developed contributes to advancing best practices used in hydrological evaluation by making available a tool that features advanced methods seldom used in the hydrological community, and by simplifying their widespread adoption by providing an efficient and easy-to-use tool that is usable in the main programming languages used by the community. We believe that, while this is not the end of the road, this represents a far from negligible step towards achieving reproducible science in hydrology and, as such, is a valuable contribution to hydrological science. However, we agree with the reviewer that this was not explained well enough in the original manuscript, therefore we have extensively reworked the introduction to emphasise these motivations and innovations around our tool (see R-L25-32, R-L50-59).

**Specific comments**

**R2C2:** The introduction would benefit from more clearly articulating what new capabilities evalhyd provides compared to existing hydrologic evaluation packages. As it stands, the motivation around standardization across languages is a bit weak.

**AR:** We modified the introduction accordingly. Please see our reply to **R2C1**.

**R2C3:** In the key functionalities section, the masking and bootstrapping methods need more detailed explanation. Pseudocode or formulas would help make these clearer.

**AR:** Thank you for raising this issue and for the suggestion. We considered using pseudo code and formulas, but we decided that figures displaying trivial examples of both functionalities would be more understandable. This is why we have produced two additional figures were produced that aim to complement the explanations for these two key functionalities (see new Figures 3 and 4 in the revised manuscript). For the sake of coherence, we took care of using similar symbology with the existing figure on the handling of missing data.

**R2C4:** For the evaluation metrics, links or references to the original sources for each metric should be provided. More justification for the specific metrics included would also help show the comprehensiveness.

**AR:** Tables 2 and 3 in the original manuscript provide the original sources, where possible. However, given the general nature (i.e. standard statistical metrics not always specific to hydrology) of some of them, references are not always possible to find.

**R2C5:** The case study, more novel demonstrations of the tool would strengthen this section.

**AR:** We do believe that the existing demonstrations of the stratification (i.e. masking) functionality or the bootstrapping functionality represent novel and advanced methods for the evaluation of hydrological predictions. The focus of our manuscript is the development of the tool and its demonstration (using data from an independent already published paper) and, given its numerous fonctionalities, additional demonstrations would take a lot of space in the paper and change its focus.

**R2C6:** The conclusions would be improved by specifically emphasizing the limitations around extensibility, visualizations, and support for continuous distributions. Comparisons to other existing packages may help contextualize the pros/cons.

**AR:** Thank you for the suggestion, this is indeed worth recapitulating the limitations as well. We have added a paragraph summarising the main limitations in the conclusions (see R-L311-312). The comparison to existing packages is already made in the paragraph before last in the introduction (see O-L32-48, or R-L33-43).

**References**

Clark, M. P., Vogel, R. M., Lamontagne, J. R., Mizukami, N., Knoben, W. J. M., Tang, G., et al. (2021). The abuse of popular performance metrics in hydrologic modeling. Water Resources Research, 57, e2020WR029001. https://doi.org/10.1029/2020WR029001.

Gneiting, T., Stanberry, L., Grimit, E., Held, L., and Johnson, N. (2008). Assessing probabilistic forecasts of multivariate quantities, with an application to ensemble predictions of surface winds, TEST, 17, 211–235, https://doi.org/10.1007/s11749-008-0114-x.

Harrigan, S., Zsoter, E., Cloke, H., Salamon, P., and Prudhomme, C. (2023). Daily ensemble river discharge reforecasts and real-time forecasts from the operational Global Flood Awareness System, Hydrology and Earth System Sciences, 27, 1–19, https://doi.org/10.5194/hess-27-1-2023.

---

## Referee Report (RR1)

Review for manuscript: *evalhyd v0.1.2: a polyglot tool for the evaluation of deterministic and probabilistic streamflow predictions*

**Summary:**

This manuscript presents a straightforward goal: to introduce an open-source package available in C++, R, Python, and cmd for computing popular metrics in the field of hydrology. The package offers a range of functionalities, including data preprocessing methods and both deterministic and probabilistic metric computation. This comprehensive approach caters to users across varying levels of modeling experience, from those seeking simplified solutions to those wishing to streamline their workflow without the need to code equations for specific metrics. Overall, this package is expected to benefit both novice users and experienced modelers alike.

**Comments:**

Line 76-77: The author highlights the capability of the C++ core of this package in handling large datasets; however, there is a lack of scalability tests or comparisons with other models to support this claim.

Table 2, Table 3: Parentheses should be used for open range indicators.

Line 91: Add "evalhyd" as the name of one of the tools.

Section 3.3: Consider using a general notation of $X^{N \times d}$ and provide examples of d as listed.

Preprocessing functionalities provided in 4.2-4.4 may be considered trivial, as other packages offer more extensive capabilities. It is suggested that the authors continue expanding this package to incorporate additional methods for broader usage.

Figure 6: Instead of plotting a 1:1 line, plotting the residual of evalhyd minus Harrigan et al. (2023) results against lead time may be more meaningful.

---

## Author Response (AR2)

Dear Editor, Dear Referees,

We would like to thank you for further considering our manuscript, and for the review by the two new referees. Please find below a detailed point-by-point answer to the referees' comments.

The authors.

**Nomenclature:**

RXCY – Referee number X Comment number Y
AR – Authors' Reply
O-LX – Original manuscript Line X
R-LX – Revised manuscript Line X

**Referee #3 (Anonymous)**

**R3C1:** Line 76-77: The author highlights the capability of the C++ core of this package in handling large datasets; however, there is a lack of scalability tests or comparisons with other models to support this claim.

**AR:** The tool has already been used within our team on operational multi-model ensemble forecasting datasets produced by the national hydrological drought forecasting online platform PREMHYCE (Nicolle et al., 2020; Tilmant et al., 2020) even though this has not lead to a publication yet. However, the tool has also been used successfully in a recent multi-model large sample study (Thébault, 2023). This reference has been added to the article (see R-L78) to support the claim. This comes as evidence that it can handle large datasets. However, we are not claiming that other packages cannot handle large datasets as well, so a comparison with other packages does not seem essential here.

**R3C2:** Table 2, Table 3: Parentheses should be used for open range indicators.

**AR:** Thank you for making us aware of this international mathematical convention, different from our local convention on open ranges. Ranges in Tables 2 and 3 have been amended in the revised manuscript to use parentheses for the relevant metric ranges.

**R3C3:** Line 91: Add "evalhyd" as the name of one of the tools.

**AR:** Thank you. We have added the name of the tool (see R-L91).

**R3C4:** Section 3.3: Consider using a general notation of XNxd and provide examples of d as listed.

**AR:** Thank you for the suggestion. Our notations were indeed not quite followinf any standard notation. We have followed your recommendation and used the more formal mathematical notation for tensors/multi-dimensional arrays (see R-L103-117).

**R3C5:** Preprocessing functionalities provided in 4.2-4.4 may be considered trivial, as other packages offer more extensive capabilities. It is suggested that the authors continue expanding this package to incorporate additional methods for broader usage.

**AR:** Yes, we agree with the reviewer that there is scope for additional functionalities, namely regarding preprocessing aspects. In particular, computing metrics on flow statistics (e.g. mean) applied on sliding windows, under a given threshold, on standardised flow indicators (e.g. QMNA [https://fr.wikipedia.org/wiki/QMNA, in French], VCN3 [https://fr.wikipedia.org/wiki/VCN3, in French] commonly used in France) could be useful. These will be considered for future versions of the tool. We have added these suggestions in the conclusions and perspectives section (see R-L317-319).

**R3C6:** Figure 6: Instead of plotting a 1:1 line, plotting the residual of evalhyd minus Harrigan et al. (2023) results against lead time may be more meaningful.

**AR:** We have tried to produce the figure with the residuals instead (see **Figure 1**) but we do not believe that these residuals are meaningful. Indeed, the results reported in Harrigan et al. (2023) only provided a precision of two decimals. If we apply a two-decimal rounding on our results with evalhyd, we obtain **Figure 2** instead. We can notice that the residuals are very often equal to zero, with only a handful of exceptions. Those exceptions are most likely due to some marginal numerical precision differences and/or rounding applied at different stages of the data processing. Therefore, we do not believe these to be meaningful and we would prefer to keep the original figure that we find more easily understandable.

[Figure]

**Figure 1:** CRPSS residuals (i.e. difference between the CRPSS reported in Harrigan et al. (2023) and the CRPSS calculated using evalhyd, applying no rounding on the latter).

[Figure]

**Figure 2:** CRPSS residuals (i.e. difference between the CRPSS reported in Harrigan et al. (2023) and the CRPSS calculated using evalhyd, applying a rounding at the second decimal on the latter to match the precision provided in the former).

**Referee #4 (Anonymous)**

**R4C1:** The tool developed in this article is an open-source tool with multiple language versions. As mentioned in the article, the tool can be used in Python, R, and C++. I would like to know if it is possible to use it in more language environments in the future, such as Java, JavaScript, etc.

**AR:** We do already touch upon the scope for more languages to be supported in the future at the end of section 3.1 (i.e. Julia and Octave). As *evalhyd* relies on the C++ core library *xtensor* for vectorised numerical computations, only Julia is reasonably close to offer an interface as the bindings between this language and C++ already exist, languages such as Octave, Java, and JavaScript would require for such bindings to be developed by the *xtensor* team or some independent contributors. While we cannot deny that there are likely Java/Javascript users in the hydrological community, these do not appear as the main languages favoured by the community in recent years. For instance, while the Ensemble Verification System (EVS) was developed by NOAA in Java (Brown et al., 2010), this was back in 2010, and more recent tools are now more often developed in Python or in R.

**R4C2:** In the introduction section, the article introduces some existing tools and introduces the shortcomings of these existing tools. I personally believe that the new tool developed in this article can be used to make a detailed comparison with existing tools, including comparing features, performance indicators, or usability, highlighting the advantages of the new tool.

**AR:** We already mention the similarities and the innovations of our tool compared to existing noteworthy tools. We agree with the reviewer that an exhaustive comparison with other existing tools would be useful. However, we believe that the choice of criteria for comparison would need to be made by a diverse group of international experts in order not to be biased towards one tool or another and establish an independent benchmark. Indeed, it is likely that if we come up with the list of criteria, despite our best efforts, we would certainly mostly consider the aspects/functionalities relevant to our own practices, and overlook others we did not think of. It is clear that all tools have their pros and cons, and they all have their best usage context. But what crucial design aspect of our tool that stands out is the polyglot character of it, making it potentially accessible to a large pool of users. The regular workshops of the HEPEX community may offer the opportunity to produce an independent benchmark.

**R4C3:** Personally, I think that nonprofessional programmers may be limited by the interactive performance of this tool. I can consider setting up some user guides or tutorials, which may make the tool easier to use by a wider audience and facilitate future expansion of the tool.

**AR:** The online documentation accessible at https://hydrogr.github.io/evalhyd/, as mentioned in the conclusions of the article, provides user guides and API references. However, we agree with the reviewer that tutorials were lacking. We have added a tutorial using the GloFAS data presented in the paper in the Python section of our online documentation. We will produce equivalent tutorials in the near future for the other *evalhyd* bindings.

**R4C4:** In the display section of the tool, a series of image results were used for display. Personally, I think it is possible to consider providing a more detailed introduction to the display results so that readers can understand the effectiveness of the tool.

**AR:** All of the figures presented in the illustrative example section are already introduced in the text to provide such introduction to the results. In addition, we believe that the captions of those figures are already quite lengthy and provide the details needed to make sure that they can be understood without relying on the main text. We are unsure how to further improve on these aspects in order to remedy this comment.

**References**

Brown, J., Demargne, J., Seo, D.-J., Liu, Y. (2010). The Ensemble Verification System (EVS): A software tool for verifying ensemble forecasts of hydrometeorological and hydrologic variables at discrete locations. Environmental Modelling & Software. 25. 854-872. https://doi.org/10.1016/j.envsoft.2010.01.009.

Harrigan, S., Zsoter, E., Cloke, H., Salamon, P., and Prudhomme, C. (2023). Daily ensemble river discharge reforecasts and real-time forecasts from the operational Global Flood Awareness System, Hydrology and Earth System Sciences, 27, 1–19, https://doi.org/10.5194/hess-27-1-2023.

Nicolle, P., et al. (2020). PREMHYCE: An operational tool for low-flow forecasting, Proc. IAHS, 383, 381–389, https://doi.org/10.5194/piahs-383-381-2020.

Tilmant, F., et al. (2020). PREMHYCE: an operational tool for low-flow forecasting, La Houille Blanche, 106:5, 37-44. https://10.1051/lhb/2020043.

Thébault, C. (2023). Quels apports d'une approche multi-modèle semi-distribuée pour la prévision des débits ? PhD thesis (in French), Sorbonne Université. https://theses.hal.science/tel-04519745.

---

## Author Response (AR3)

Dear Editor,

Thank you for your suggestions.

In this revised submission, we have shared and archived the HTML documentation for `evalhyd` on *Software Heritage* (which gives a permanent identifier called a SWHID, acting similarly as a DOI). This HTML documentation corresponds to the user guide for the entire `evalhyd` software stack and, in particular, it contains tutorials in Python and in R using a sample of the same dataset as the one used for the illustrative example in the article.

In addition, since you mentioned increasing the repeatability (i.e. reproducibility) of our work, we carefully prepared and shared all the Python scripts that were used to produce the scientific figures presented in the article on *Zenodo*. We made sure for the scripts to be well commented throughout so as to provide additional training material for the reader, in complement to the user guide and tutorials.

As requested, these two additional resources are mentioned in the "Data availability" section of the newly revised article.

The authors